# Attributing the Impacts of Vegetation and Climate Changes on the Spatial Heterogeneity of Terrestrial Water Storage over the Tibetan Plateau

Yuna Han [1], Depeng Zuo [1,*], Zongxue Xu [1], Guoqing Wang [2,3], Dingzhi Peng [1], Bo Pang [1] and Hong Yang [4]

1 Beijing Key Laboratory of Urban Water Cycle and Sponge City Technology, College of Water Science, Beijing Normal University, Beijing 100875, China
2 State Key Laboratory of Hydrology-Water Resources and Hydraulic Engineering, Nanjing Hydraulic Research Institute, Nanjing 210098, China
3 Research Center for Climate Change, Ministry of Water Resources, Nanjing 210029, China
4 Eawag, Swiss Federal Institute of Aquatic Science and Technology, 8600 Dübendorf, Switzerland
* Correspondence: dpzuo@bnu.edu.cn; Tel.: +86-10-58801136

**Abstract:** Terrestrial water storage (TWS) is of great importance to the global water and energy budget, which modulates the hydrological cycle and then determines the spatiotemporal distributions of water resources availability. The Tibetan Plateau is the birthplace of the Yangtze, Yellow, and Lancang–Mekong River, where the water resources are directly related to the life of the Eastern and Southeastern Asian people. Based on multi-source datasets during the period 1981–2015, the long-term spatiotemporal variabilities of the TWS over the Tibetan Plateau were investigated by the Sen's slope and Mann–Kendall test trend analysis methods; the changing mechanisms were explored from two perspectives of components analysis and the hydrological cycle. The water conservation capacity of vegetation in the alpine mountainous areas was also discussed by geostatistical methods such as correlation analysis, extracted by attributes and zonal statistics. The results show that the TWS of the Tibetan Plateau increased with the speed of 0.7 mm/yr as the precipitation accumulated and the glaciers melted during the period 1981–2015. The TWS values were low and generally present a trend of obvious accumulation over the northern Tibetan Plateau, while the high and decreasing values were distributed in the south of Tibetan Plateau. The results of the components analysis indicate that the TWS mainly consisted of soil moisture at one-fourth layers, which are 0–200 cm underground in most areas of the Tibetan Plateau. The precipitation is mainly lost through evapotranspiration over the northern Tibetan Plateau, while in the northwestern corner of the Tibetan Plateau, the Himalayas, and northeastern Yarlung Zangbo River basin, the runoff coefficients were larger than 1.0 due to the influence of snow melting. In the alpine mountains, different climate and vegetation conditions have complex effects on water resources. The results are helpful for understanding the changing mechanism of water storage over the Tibetan Plateau and have scientific meaning for the development, utilization, and protection of regional water resources.

**Keywords:** terrestrial water storage (TWS); spatiotemporal variabilities; components analysis; hydrological cycle; Tibetan Plateau





## 1. Introduction

Climate change and its impacts on water resources is a major issue which China and many other countries are facing or will have to cope with in the future [1,2]. China is a severely water-deficient country, occupying only 7.7% of the global freshwater resources, whilst having 18.5% of the world' population [3]. Meanwhile, the unevenly distributed climate and ecosystems in China also reveal a greatly spatial heterogeneity of water resources between the south, where water is abundant, and the drier north. Over the past several decades, China has already experienced some devastating climate extremes which

caused immeasurable economic loss [1], such as the great flood of 1998 in the Yangtze River Basin, the snow disaster of 2008 in South China and the severe drought of 2010 in southwest China. As climate change and explosive economic growth have intensified, the ecological environment and water resources in China have undergone severe challenges, which may cause China's economic growth to be vulnerable to climate change and resource constraints [4–6].

Tibetan Plateau, known as the "Asian Water Tower" and "the Third Pole" because of the abundant water resources and it having the highest elevation in the world, is the birthplace of more than a dozen water systems, and in which the rivers provide water resources for more than two billion Asian people, including the Chinese [7–9]. Both the "Climate change 2021: the physical science basis" assessment report [2] and the "Blue book on climate change in China 2021" [10] point out that the global temperatures and sea levels are still rising, the Arctic Sea ice area is reducing, and other issues also remain serious. The alpine regions are highly sensitive and prone to climate change, and the warming rates are twice as much as the global average [11]. Given the unique geographical location and climatic conditions, the ecosystem of the Tibetan Plateau is very fragile [12,13]. In the context of the warming climate, permafrost degradation, permanent glacier and snow melting, lake expansion and other phenomena happening on the Tibetan Plateau have significantly transformed its hydrological processes [14] and posed severe challenges to local, ecological, environmental protection and water resources management. Understanding how climate change affects the plateau's ecosystem and how both vegetation and climate changes impact on the spatial heterogeneity of water resources that are stored in a gaseous, liquid, and solid state over the Tibetan Plateau are crucial for water security and Asian life.

As the critical component of the water and energy budget, terrestrial water storage (TWS) includes water stored in rivers, lakes, reservoirs, snow, glaciers, plant canopies, soil, and underground [15], which can modulate the hydrological cycle and then determine the availability of water resources, and it plays a critical role in the changing earth system [16–20]. Scholars have carried out a series of research on the characteristics of TWS at various temporal and spatial scales. Wang et al integrated satellite and modeling data, analyzed the TWS changes of endorheic and exoreic river basins globally during the period of 2002–2016, and recommended monitoring long-term variations [21]. Pokhrel et al examined the future change in global TWS during the period 2030–2099 and the linkages to drought by using ensemble hydrological simulations based on the historical baseline period 1976–2005 [17]. At the same time, some studies focused on the regional scale to explore the variabilities of the TWS and their corresponding attributions [14,22,23]. TWS in the Yellow River source region during the period 2003–2015 increased corresponding to the increase in runoff and soil moisture, whereas it depleted in the midstream and downstream due to the decreasing runoff and groundwater [23]. The fluctuations in soil moisture and the glacier are the primary reasons for the TWS changes in several river basins over the Tibetan Plateau [14].

The traditional methods to estimate the characteristics and variabilities of the TWS include in situ observations and hydrological modeling, which are limited to the point scale or the uncertainties in models [24,25]. In addition, high-quality spatiotemporal information of the water balance component is difficult to obtain due to the lack of global in situ measurements [26]. Therefore, remote sensing datasets offer an excellent observation approach and have been able to quantitatively assess large-scale and long-period hydrological cycle variables with the development of "3S", which is the shorter form of remote Sensing (RS), geography information systems (GIS), and global positioning systems (GPS). Especially in the high-altitude regions, lacking in situ observations greatly limits the awareness in the TWS changes over the Tibetan Plateau. The mission of the gravity recovery and climate experiment (GRACE) satellite has provided convenient data sources for calculating the TWS in a large-scale region since March 2002 [27–29]. Muskett and Moiwo et al detected an overall reduction in TWS at the whole Tibetan Plateau scale during the periods of 2002–2006 and 2003–2008, respectively [30,31]. Jiao et al and Xiang et al found that a decreasing trend

of the TWS in the south and an incremental trend in the inner and northern Tibetan Plateau during the periods 2003–2009 and 2003–2012 [32,33]. Nevertheless, previous research has mostly focused on the Tibetan Plateau as a whole plateau and neglected the spatial heterogeneity of the TWS changes. In addition, current studies have mainly addressed the inter-annual changes in TWS and mostly lack the characteristics of intra-annual variability in the TWS and the driving mechanism over the Tibetan Plateau.

Changes in the climate and the underlying surface, especially for the vegetation types and coverage, could significantly alter hydrological processes such as precipitation, evapotranspiration, runoff generation, and concentration in a watershed, which feeds back into the production and life of residents [34,35]. The increased precipitation may promote the growth of vegetation, thus improving the efficiency of water conservation, whereas the rise in temperature may lead to an increase in evapotranspiration and then a decrease in TWS. Lorenz et al compared various precipitation, evapotranspiration, and TWS datasets at a global scale [36]. Choosing which combination is the best one is hard; evapotranspiration and water storage were the basis of a stable hydrological cycle for most river basins. Yang and Chen summed up the soil moisture, snow-equivalent water, and canopy water from the global land surface data assimilation system (GLDAS) and calculated the TWS the same as the GRACE; the results showed that the correlation coefficient of TWS between the two methods was 0.75, with it being statistically significant [37]. Weber et al compared the applicability of various global datasets in ungauged alpine basins; to some extent, the precipitation, temperature, and snow cover derived from different versions of GLDAS were quite reasonable against the measurements [38]. Otherwise, the in-situ soil moisture observations of 16 sites in the ungauged alpine basins also confirmed the consistency of GLDAS in the expression of trend changes [39]. Moreover, because the satellite data assimilation provided relatively reliable soil moisture estimates, the TWS reconstructed by the water balance equation is more accurate than the long-term or monthly mean of GRACE [26,40].

The current understanding does not allow for a clear assessment for the impacts of vegetation and climate changes on the spatial heterogeneity of TWS at the long-term series and sub-regional scales over the Tibetan Plateau. Therefore, based on the global land surface data assimilation system (GLDAS), China meteorological forcing dataset (CMFD), and the third-generation global inventory modeling and mapping studies normalized vegetation index (GIMMS NDVI3g) during the period 1981–2015, this study aims to reveal the decadal spatio-temporal variabilities of climate, vegetation, and TWS trends to explore the driving mechanism of changes in the TWS and associate TWS with likely future water resources and environment protection over the Tibetan Plateau. The specific objectives include that: (1) the trend analysis methods were used to explore the long-term spatio-temporal variabilities including the spatial heterogeneity of the TWS; (2) the water balance method, associated with geostatistical analysis, such as the correlation analysis, and extracted by attributes and zonal statistics, were adopted to conduct the components analysis and identify the mechanism of changes in the TWS under the influences of the hydrological cycle; and (3) in addition, the influences of climate change on water resources and the water conservation capacity of vegetation were also discussed in the twelve sub-regions over the Tibetan Plateau. The results can help to deeply understand the spatiotemporal characteristics and mechanisms of change in the TWS over the Tibetan Plateau, which can provide a reference for water resource development, protection, and management in East Asia, Southeast Asia, and other regions.

## 2. Study Area and Data Description

### 2.1. Study Area

The Tibetan Plateau (26°00′12″~39°46′50″N, 73°18′52″~104°46′59″E) has the highest altitude in an world with the average elevation of 4320 m [13] and is called as the "roof of the world". The plateau is 1560 km long from north to south and 3360 km wide from west to east, with an overall area of more than $2.5 \times 10$ km$^2$ [41], including the provinces

of Tibet and Qinghai, as well as parts of Xinjiang, Gansu, Sichuan, and Yunnan, with the total population of 13.74 million. The terrain over the Tibetan plateau is complex, mainly including basins, mountains and valleys (Figure 1), which provide good conditions for storing water resources and are the source regions of great rivers, such as the Yangtze River, Yellow River, and Lancang–Mekong River [7]. Because of the great differences in elevation and the steep slope, there are many canyons in the outflow river system. The typical plateau continental climate over the Tibetan Plateau alternates between hot and cold temperature, distinguishes dry and wet humidity, and has small annual temperature difference, long hours of sunlight, and intense radiation. In the summer, the prevailing southeast, south, and southwest monsoons transport a lot of water vapor to the plateau, while the westerly wind circulation is dominant in the winter, making precipitation significantly decrease [41]. The average annual precipitation is about 400 mm, which is about 30 mm in the northwestern Tibetan Plateau, but it exceeds 2000 mm in the southeastern regions, and it is characterized as gradually increasing from north to south [42]. The average air temperature is −6~20 °C and is generally similar to the spatial characteristics of precipitation [43], which is significantly higher in the eastern monsoon region than that in the western region. Glaciers, snow, and frozen soil are widely distributed over the Tibetan Plateau, especially the perennial and seasonal permafrost due to the cold climate.

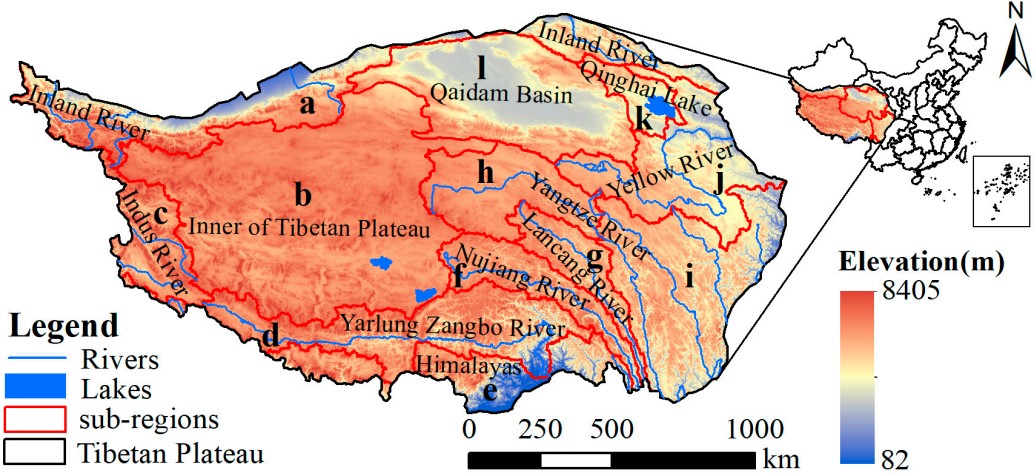

**Figure 1.** Schematic diagram of the geographic location of the Tibetan Plateau.

There are 57 soil types and 141 soil subtypes across the Tibetan Plateau, among which tierra is the main soil type. Frigid calcic soil and frigid frozen soil are mainly distributed in the inner of the Tibetan Plateau (Figure 2), accounting for 27.01% and 10.41% of the whole plateau, respectively. In the southeast of the plateau, felty soil, dark felty Soil, Lixisols and Ferralsols are unevenly distributed [44]. The greatest parts of the vegetation types are the alpine grassland and natural cover, with shrubland, needleleaf, and broadleaf distributed in the southeast, which are related to the spatial distributions of the climate and soil [45]. A figure of 74.79% of the whole Tibetan Plateau is a moderately fragile ecological environment [46], where vegetation plays a key role in ecosystem protection and water conservation. Given the comprehensive consideration of natural conditions, Chinese watershed distributions, administrative divisions, and the "one belt, one road boundary map of key basins in Asia" [47], the Tibetan Plateau was divided into 12 sub-regions for conducting an in-depth study from the whole basin and the regional perspectives.

### 2.2. Data Description

#### 2.2.1. Global Land Data Assimilation System (GLDAS)

The GLDAS was developed to generate optimal fields of land surface states and fluxes by ingesting satellite- and ground-based observational data products, using advanced land surface modeling and data assimilation techniques [48] and it can be extracted from the Goddard Earth Science Data and Information Service Center (https://disc.gsfc.nasa.gov/,

accessed on 10 October 2021). Currently, GLDAS drives the Noah, mosaic, community land model (CLM) and variable infiltration capacity (VIC), four land surface models, which consist of different components or indices. The Noah model has more complete observation elements, with 39 different variables such as snow water equivalent, soil moisture data at one-fourth layers (10, 40, 100, and 200 cm underground) and plant canopy surface water, etc. It also has the longer time series from 1948 to present, more time intervals of 3 h, daily, and monthly, as well as the finer spatial resolution of 0.25° and 1° among the four models.

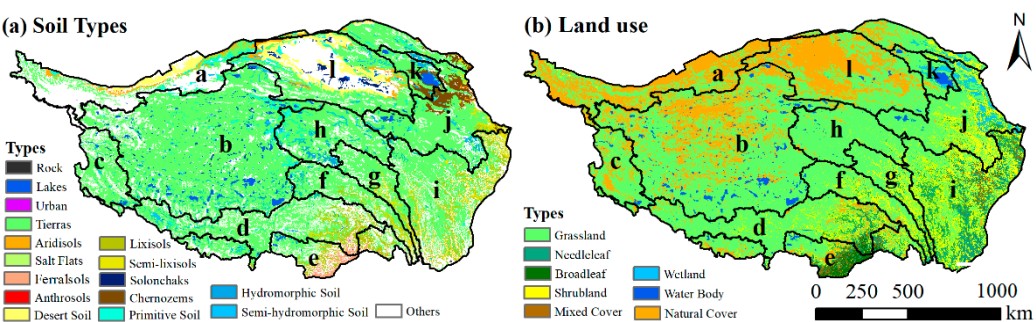

**Figure 2.** Spatial distributions of soil and land use types over the Tibetan Plateau.

### 2.2.2. China Meteorological Forcing Dataset (CMFD)

Previous studies generally considered the unevenly distributed site observation data to represent the spatiotemporal characteristics of climate, which is not sufficient to indicate the spatial heterogeneity. The precipitation and temperature are important factors indicating long-term climate change and are collected from the CMFD (http://data.tpdc.ac.cn/, accessed on 20 November 2021), which includes seven elements of near-surface air temperature, precipitation rate, wind speed, humidity, pressure, shortwave radiation, and longwave radiation, and it has the shortest time interval of 3 h and highest spatial resolution of 0.1°. The CMFD is comprised remote sensing, radar, and site observation data from 1979 to 2018 across China. Additionally, some studies have substantiated the accuracy and reliability of application in alpine regions [49–51].

### 2.2.3. The Third-Generation Global Inventory Modeling and Mapping Studies Normalized Difference Vegetation Index (GIMMS NDVI3g)

NDVI can reflect the spatial variability of vegetation cover and also enumerates the impact of the environment on vegetation [52,53]. Due to the simple estimation, easy availability at different spatial and temporal resolutions, and cancellation of noise that is caused due to the solar angle, topographic illumination, clouds, and atmospheric conditions, NDVI has been used in many relevant pieces of research [52,54] and in the current study. The GIMMS NDVI3g dataset was used to express the vegetation specialties during the period 1981–2015. The advanced very high-resolution radiometer (AVHRR) is used to collect data, the spatial resolution is 0.083°, and has an interval of half-month (https://ecocast.arc.nasa.gov/data/pub/gimms/, accessed on 2 February 2020). To minimize the disadvantageous influences, the data have been carefully corrected [55] and have a relatively high quality [49]. Furthermore, the maximum value composite method based on the two NDVI images of each month was adopted, and then monthly data were developed to reduce the interferences of other elements. The GIMMS NDVI3g is deemed as the uncommonly available dataset for the long-term vegetation variabilities research.

### 2.2.4. Other Datasets

The datasets of the study area boundary, watershed distribution, digital elevation model (dem), the soil database of china, and land cover classification were freely obtained from the Resource and Environmental Science Data Center of the Chinese Academy of Sciences, the National Tibetan Plateau Data Center, the Institute of Soil Science, the Chinese Academy of Sciences, and the University of Maryland, respectively. Detailed information

is shown in Table 1. Among them, the study area boundary is clipped from the China Municipal Administrative Zoning Map in 2019, the watershed distribution is collected from the "one belt, one road boundary map of key basins in Asia" [47], and the 1-km DEM, soil types, and land use data can show the basic characteristics over the Tibetan Plateau, which all are essential to guide the division of sub-regions.

**Table 1.** The detailed information of all the datasets in the current study.

| Type | Data | Description | Source | Postscript |
|---|---|---|---|---|
| Meteorological data | Global Land Data Assimilation System (GLDAS) | 1948–present, 0.25°, 1° | Goddard Earth Science Data and Information Service Center | https://disc.gsfc.nasa.gov/ (accessed on 10 October 2021). |
| | China Meteorological Forcing Dataset (CMFD) | 1979–2018, 0.1° | National Tibetan Plateau Data Center | http://data.tpdc.ac.cn (accessed on 20 November 2021). |
| Underlying surface data | Digital Elevation Model | 2000, 90 m | Resource and Environmental Science Data Center of the Chinese Academy of Sciences | https://www.resdc.cn/ (accessed on15 January 2020). |
| | Soil Database of China | 2002, 1:1,000,000 | Institute of Soil Science, Chinese Academy of Sciences | http://www.issas.ac.cn/ (accessed on 5 December 2022). |
| | Land Cover Classification | 1998, 1 km | University of Maryland | http://www.landcover.org/data/landcover/data.shtml (accessed on 5 December 2022). |
| | GIMMS NDVI3g | 1981–2015, 0.083° | National Aeronautics and Space Administration | https://ecocast.arc.nasa.gov/data/pub/gimms/ (accessed on 2 February 2020). |

## 3. Methodology

Given the differences in initial format, the spatial and temporal resolutions of the datasets, all the precipitation, temperature, NDVI, and TWS components during the period 1981–2015 were extracted from the GLDAS, CMFD, and GIMMS NDVI3g datasets, respectively, which are stored information in the network common data form (NetCDF) files [53,56,57]. Then, temporal and spatial interpolation techniques were carried out to deal with the datasets, including reprojection and resample using the data management tool at 0.1°, which were converted into monthly/annual data using the time weighting method [58]. Subsequent raster calculations and a critical description of the methodologies (Table 2) are shown as follows.

**Table 2.** Critical description of the methodologies in the current study.

| Method | Description |
|---|---|
| Calculation of the terrestrial water storage (TWS) | Based on the water balance equation, the GLDAS was adopted to calculate the TWS and further determinate the water resources over the Tibetan Plateau. It can provide the basic information for the characteristic analysis and the changing mechanism of the TWS and its components under the influences of hydrological cycle. |
| Sen's slope and Mann–Kendall trend test | Appling the non-parametric methods investigated the spatiotemporal variabilities of the TWS and its different components, climatic elements, and vegetation cover over the Tibetan Plateau from 1981 to 2015. |
| Geostatistical methods | To identify the changing mechanism of TWS and the water conservation capacity of vegetation in the sub-regions scale over the Tibetan Plateau, the Pearson correlation analysis, extracted by attributes and zonal statistics, etc., was used to filter and analyze the data. |

### 3.1. Calculation of the Terrestrial Water Storage

The TWS contains all sources of water, including snow, glaciers, soil moisture, and biological water, etc. The Noah model in the GLDAS provides the soil moisture, snow water equivalent, and plant canopy surface water data, and then the TWS was calculated

by summing up the total elements based on the water balance method during the period 1981–2015. The calculation formula is as follows:

$$TWS = Canopy + \sum_{i=1}^{4} Soil_i + Snow \tag{1}$$

in which, $TWS$ is the terrestrial water storage (mm), $Canopy$ is the plant canopy surface water (mm), $Soil_i$ is the soil moisture of the layer $i$ (mm), and $Snow$ is the snow water equivalent (mm).

### 3.2. Sen's Slope and Mann–Kendall Trend Test

Both the Sen's slope and the Mann–Kendall trend test (MK) were non-parametric methods, which were applied to investigate the spatiotemporal variabilities of the climate, vegetation, TWS, and its different components for a long-term period in this study. To express the significance level of sample changes, the MK was developed by Mann [59] and Kendall [60] and was gradually extended fields to climatology studies. Up until now, the MK has also widely applied in the detections of hydrological variables due to its advantages of low sample distribution requirements.

The Sen's slope used the median of slope ($Q_{med}$) to show the trend steepness [61] and is always applied along with the MK; the specific formulas can refer to the previous pieces of research [62,63]. The classification of the changing trends and their significance are shown in Table 3.

**Table 3.** Classification of the changing trends and their significances.

| Changing Trend | $p$ Value | Significance |
|:---:|:---:|:---:|
| slope > 0 | $p < 0.01$ | extremely significant increase |
| | $0.01 < p < 0.05$ | significant increase |
| | $0.05 < p < 0.1$ | weak significant increase |
| | $p > 0.1$ | non-significant increase |
| slope < 0 | $p < 0.01$ | extremely significant decrease |
| | $0.01 < p < 0.05$ | significant decrease |
| | $0.05 < p < 0.1$ | weak significant decrease |
| | $p > 0.1$ | non-significant decrease |

### 3.3. Geostatistical Methods

Geostatistical methods, such as Pearson correlation analysis, extracted by attributes and zonal statistics, were used to filter and analyze the data in this study. Correlation coefficients among different elements can effectively express the positive or negative relationships and the degree of correlation [64]. The relationships among the water resources, climate, and vegetation presented the probable impacts of climate change on water resource and the water conservation capacity of vegetation at a certain degree. Extracting and counting data were mostly executed in the software of ArcGIS and the correlation analysis was calculated by the following equation:

$$R = \frac{\sum_{i=1}^{n}(x_i - \overline{x})(y_i - \overline{y})}{\sqrt{\sum_{i=1}^{n}(x_i - \overline{x})^2}\sqrt{\sum_{i=1}^{n}(y_i - \overline{y})^2}} i = 1, 2, 3 \dots n \tag{2}$$

in which, $R$ is the correlation coefficient of two elements and $\overline{x}$ and $\overline{y}$ are the average values of the two elements in the time series.

## 4. Results and Analysis

### 4.1. Spatiotemporal Heterogeneity of the TWS over the Tibetan Plateau

4.1.1. Interannual Variations of the TWS

The spatial distribution and its changing trend in the TWS over the Tibetan Plateau during the period 1981–2015 are demonstrated in Figures 3 and 4. The multi-year average TWS overall increased from north to south across the Tibetan Plateau (Figure 3a), in which the minimum value of 256 mm was distributed in the northeast Qinghai Lake region (k) and the surroundings of the Qilian Mountains, while the maximum reached 52,236 mm around Mount Everest. In more than 90% of the total Tibetan Plateau, the TWS was between 300 and 600 mm. Only about 0.6% of the Tibetan Plateau produced TWS above 800 mm, which was mostly distributed near the southern Himalayas (e). The TWS in the northern Tibetan Plateau, such as the Inland River region (a), Qaidam Basin (l), Qinghai Lake region (k), the source and partly upper regions of the Yellow River (j), and Yangtze River (h), were relatively low and mainly ranged below 500 mm. Especially for the Qilian Mountains and the Qinghai Lake region (k) in the northeastern plateau, the TWS was less than 300 mm, whereas in the southern Tibetan Plateau, including the basins of the Indus River (c), Yarlung Zangbo River (d), Nujiang River (f), Lancang River (g), and the Inner of Plateau (b), the TWS was more than 500 mm. The regions with extremely high TWS were mainly located in the mountain area, where they are covered by snow and glaciers and the value could exceed 50,000 mm.

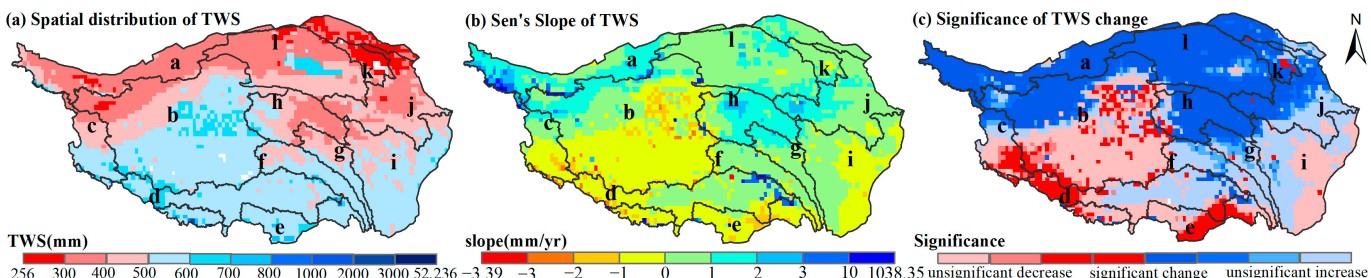

**Figure 3.** Spatial distribution and its changing trend of terrestrial water storage over the Tibetan Plateau during the period 1981–2015.

The changing trend of the TWS over the Tibetan Plateau was classified to eight ranges according to the slope and the *p* value of the significance test (Table 3 and Figure 3c). The TWS shows an extremely significant increase in the majority of the northern and eastern Tibetan Plateau during the period 1981–2015, with a decrease in the southern Tibetan Plateau, especially for the Indus River (c) and Yarlung Zangbo River source regions (d). The areas with relatively low TWS in the Tibetan Plateau have shown an extremely significant increase and indicate a larger increase in the Inland River region (a), the Yangtze River (h), the Yellow River (j), and the Lancang River source regions (g). The average slope of the TWS in the Inland River region (a) was about 2.12 mm/yr and can regionally reach to 20 mm/yr, which was mainly represented by the increases in the Karakoram Mountains, while the TWS of higher value regions in the west and south of the Tibetan Plateau have a decreasing trend, and the rate of descent can reach 3.39 mm/yr. Affected by the warm and humid Indian Ocean current, the increasing water vapor, accumulating snow and glaciers, and rising soil moisture make the TWS increase in the Nyainqen-Tanglha Mountains around the northeastern Yarlung Zangbo River Basin and the vicinities of the Himalayas. With a large base of more than 800 mm, the TWS is highly volatile and the maximum growth rate approaches 1038.35 mm/yr around Mount Everest.

The value of the annual TWS and its standard deviation for each sub-region during the period 1981–2015 were extracted and the multi-year average TWS was also calculated. The spatial distribution of multi-year average TWS was shown in the center of Figure 4, which ranged from 324.45 mm of the Qinghai Lake region (k) to 663.17 mm in the Himalayas (e). The outer circle was the long-term trend lines of the TWS in 12 sub-regions from

1981 to 2015. From the figure it can found that the standard deviation of the TWS in the Inland River region (a) was less than 50 mm before 2010, the TWS was maintained at about 325 mm, and the volatility was low. In addition, then there was a sharp increase from 2010 to 2011, and the volatility increased accordingly. The TWS in the Qiang-tang Plateau (b), Qaidam Basin (l), Yarlung Zangbo River Basin (d), and Qinghai Lake region (k) firstly rose and then showed different degrees of decline from 2010 to 2015. While for the Indus River Basin (c), the Yangtze River (h), and Yellow River (j) source regions, and the Nujiang River (f) and the Lancang River (g) basins, the TWS firstly decreased and then increased during the period 1981–2015, and the high value ultimately emerged in 2015. The standard deviation of the TWS can reach more than 1000 mm around the Himalayas due to the large base and the melting of snow and glaciers.

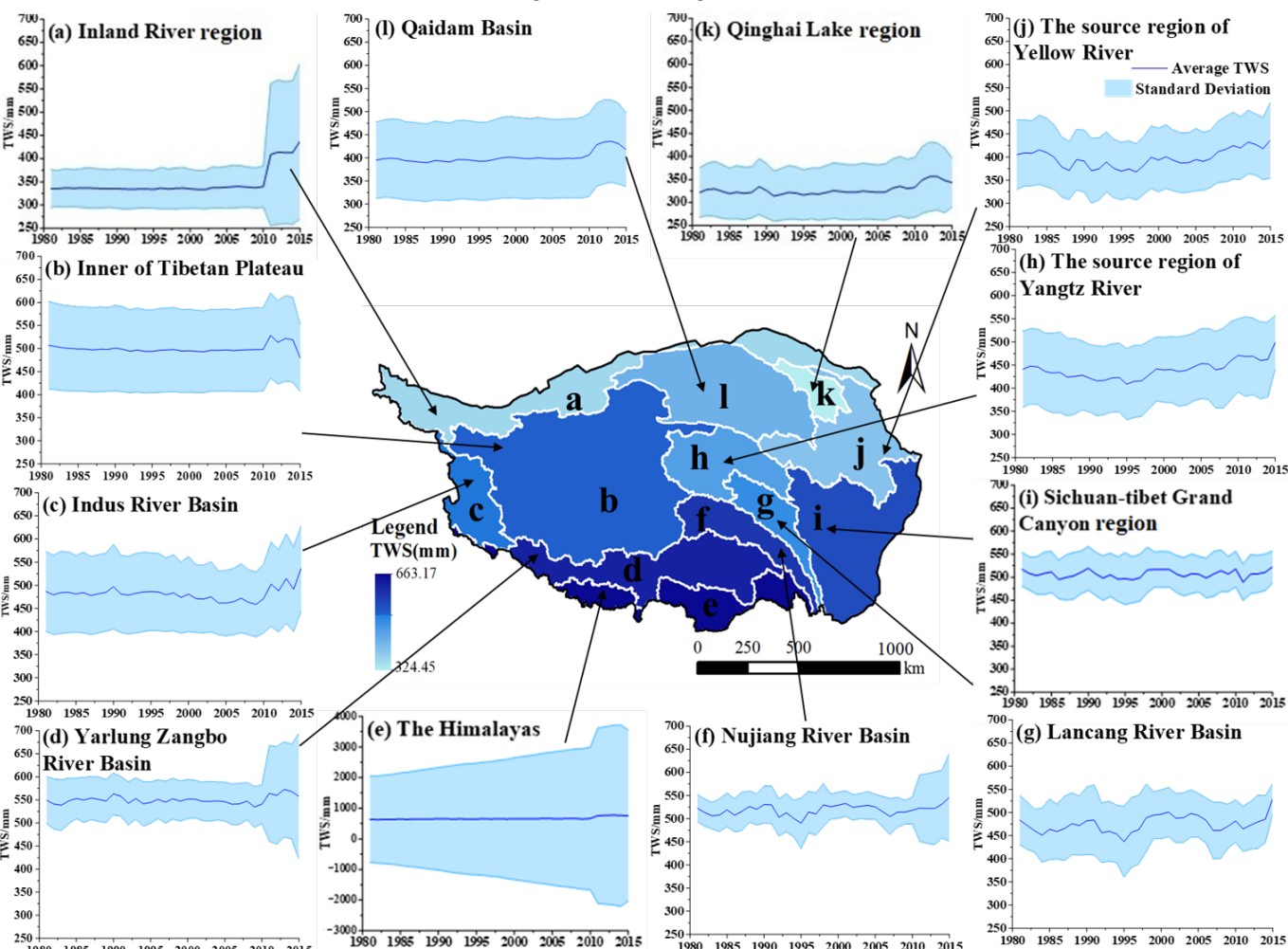

**Figure 4.** Temporal trend in terrestrial water storage for each sub-region over the Tibetan Plateau during the period 1981–2015.

### 4.1.2. Intra-Annual Variations of the TWS

The TWS presented obvious periodic changes from January to December and had significant spatial heterogeneity (Figure 5). The values remained relatively low in winter and spring and increased overall from summer to autumn. Affected by the eastern monsoon climate, the precipitation of the Southern Tibetan Plateau was unevenly distributed, with less precipitation in winter and spring, but more than 70% of the annual precipitation in summer and autumn, which makes the TWS change periodically. In the Southern Tibetan Plateau, the TWS was substantially maintained around 500 mm from January to May and then continued to increase to above 560 mm, reaching the maximum in August, and gradually falling to 500 mm from August to December. The north regions were mainly

influenced by the west wind system under the barrier of the mountains, where there are no obvious differences in weather conditions, and the TWS remained basically stable. The areas with a low TWS, such as the Inland River region (a), Qaidam Basin (l), Qinghai Lake region (k), the Yellow River (j), and the Yangtze River (h) source regions in the northern plateau, had a basically constant TWS with small fluctuations less than 20 mm and a maximum value in July.

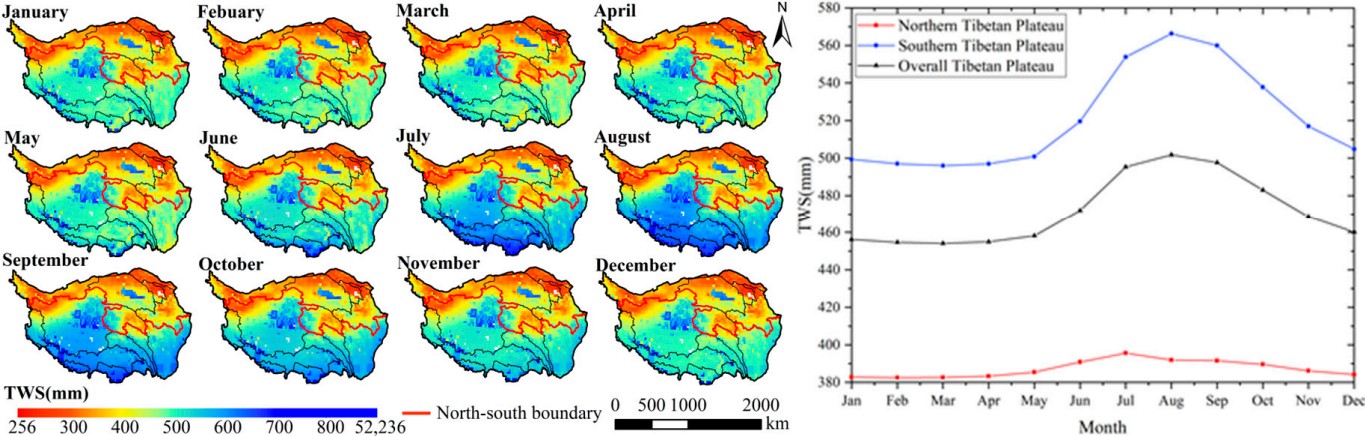

**Figure 5.** Spatial distributions of intra-annual terrestrial water storage over the Tibetan Plateau.

*4.2. Components Analysis of the TWS over the Tibetan Plateau*

4.2.1. Spatial Distributions of Each Component

Including the plant canopy surface water, the soil moisture of four different depths (10, 40, 100, and 200 cm underground), and the snow water equivalents, the proportions of each component in the TWS over the Tibetan Plateau are demonstrated in Figure 6. On the whole, soil moisture was the dominating component in TWS, accounting for more than 90% of the gross TWS in most areas. In addition, the proportion of soil moisture under the depth of 100–200 cm in the TWS is more than 70% in some regions of the Qiang-tang Plateau and the Qilian Mountains in the northeast of the Tibetan Plateau. Affected by geographical location and topographical conditions, the distributed vegetation types of the Tibetan Plateau are mainly alpine meadows, steppe, and sparse grasslands; the ecological environment is fragile. The plant canopy surface water only accounted for about 0–0.04% of the TWS, which gradually increased from northwest to southeast. The overall trend in plant canopy surface water was consistent with that in vegetation growth, and the proportion of broad-leaved forests in the tropical monsoon climate zone around the southeastern region of Tibetan Plateau was slightly higher than that of other regions. The distributions of soil moisture at different depths revealed significantly spatial heterogeneity. The northern part of the Qiang-tang Plateau had very low soil moisture at 0–100 cm, while it had a high value at the depth of 100–200 cm. In addition, then, in the Shule River, Heihe River, and Shiyang River basins in the northeast Inland River region (a) and the Qinghai Lake region (k), the soil moisture at the surface of 0–10 cm and the 100–200 cm deep layer were richer, with a poor proportion in the middle layer of 0–100 cm. A relatively high proportion of soil moisture at 0–40 cm appeared at the three rivers source regions, and the proportions were low in the deep layers, which may be decided by the influence of different vegetation types in root water-holding capacity around the different regions. High snow water equivalent areas appeared in the Karakoram Mountains in the northwestern Tibetan Plateau, the southern Himalayas (e), and partial Indus River Basin (c). Especially for the regions near Mount Everest, the proportion of snow water equivalent in the TWS can reach 98.22%.

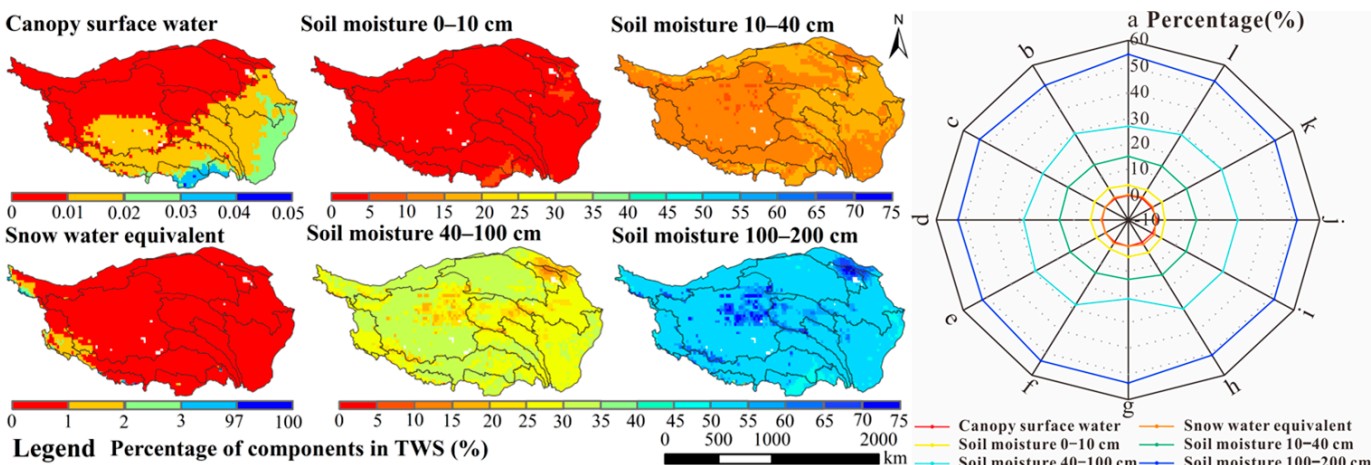

**Figure 6.** Spatial distributions of the proportions of the terrestrial water storage components over the Tibetan Plateau.

### 4.2.2. Spatiotemporal Variabilities of Each Component

To investigate the possible reasons for the TWS change based on the slope of each component during the period 1981–2015, including the plant canopy surface, soil moisture at different layers, and snow water equivalent, the trend analysis method was used in this study (Figure 7). Except for the decrease in the Indus River source region and the southwestern Yarlung Zangbo River Basin and the tropical monsoon climate zone in the southeast of the Tibetan Plateau, the plant canopy surface water in most areas showed an increasing trend, which was most obvious in the lower reaches of Yarlung Zangbo River because of the improving vegetation coverage [65]. The soil moisture at each layer within 0–200 cm showed the same changing trend, which was similar to the TWS change, increasing in the northern and eastern regions and decreasing in the western and southern Tibetan Plateau. The most significantly increasing trend occurred in the Karakoram Mountains and Tanggula Mountains, while the soil moisture at different layers in the tropical monsoon climate zone showed a clear reduction especially for the depth of 0–100 cm. Finally, for the snow water equivalent, only some regions around the southern Himalayas showed a significant increasing trend, and due to the large TWS base, the speed of acceleration can reach 1034.94 mm/yr. Therefore, soil moisture is a decisive component in the TWS over the Tibetan Plateau due to its extremely high proportion, which is inseparable from local ecological environment protection and the role of vegetation roots in water and soil conservation.

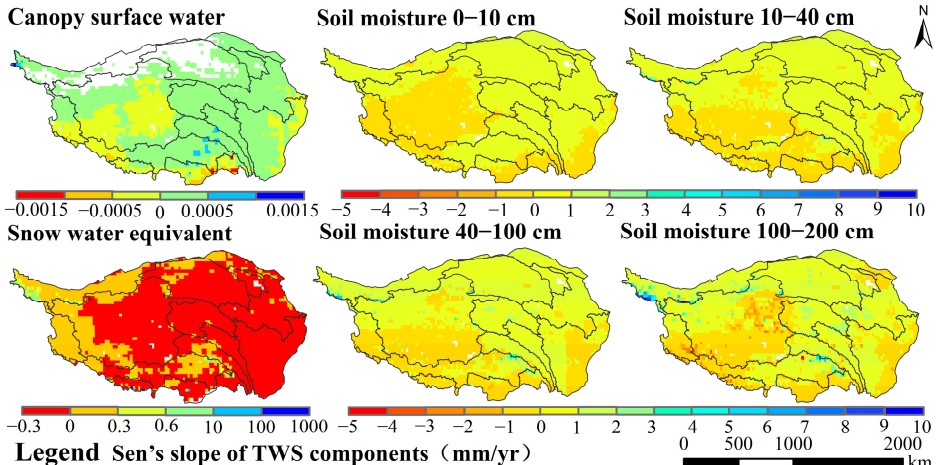

**Figure 7.** Spatial distributions of the trends in the terrestrial water storage components over the Tibetan Plateau.

*4.3. Impacts of Hydrological Cycle on the TWS*

4.3.1. Characteristics of the Hydrological Cycle Components

      Global climate change and underlying surface change interfere with water resources by affecting the precipitation, evapotranspiration, and runoff in the hydrological cycle, which are often regarded as the dominant driving factors of TWS change. The multi-year average precipitation, evapotranspiration, runoff, and TWS over the Tibetan Plateau were calculated in this study (Figure 8). They all had obviously spatial heterogeneity and gradually increased from north to south. The precipitation was below 800 mm and the evapotranspiration was below 500 mm in most areas of the Tibetan Plateau, while in the southeastern Yarlung Zangbo River Basin (d) and the Himalayas (e), the maximum precipitation was about 2000 mm and evaporation was close to 1000 mm. After deducting evaporation, most of the precipitation was formed as runoff through the hydrological processes. Influenced by the climate and topography, the southern part of the Tibetan Plateau is abundant in precipitation, and the rivers are mainly outflowing rivers with relatively large runoff. While the northern areas are blocked by high mountains, the warm air currents are difficult to reach and precipitation is scarce; snow melting and glacial replenishments are the important sources of runoff.

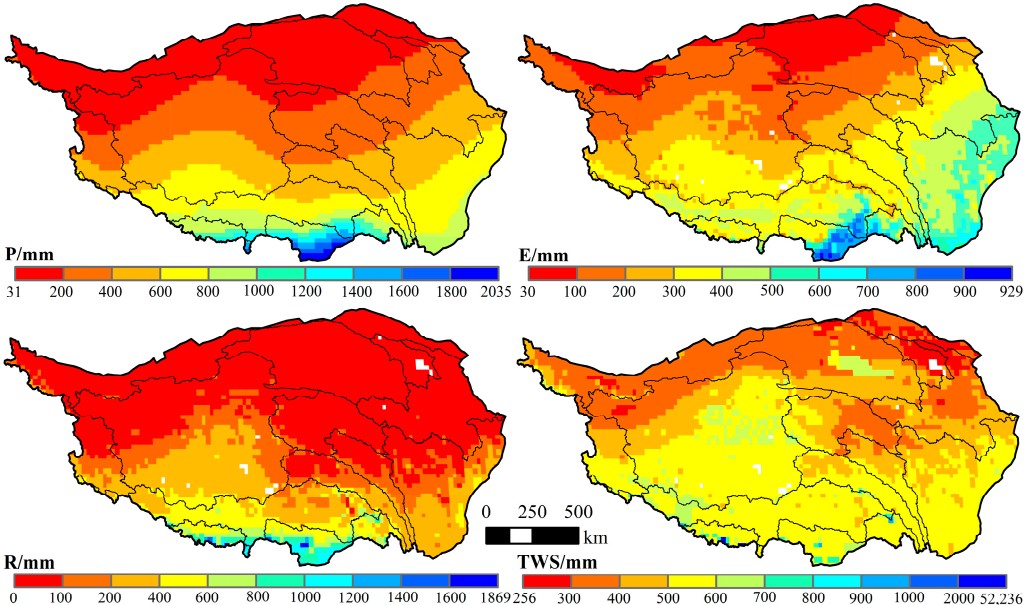

**Figure 8.** Spatial distributions of hydrological cycle's elements and terrestrial water storage changes over the Tibetan Plateau.

4.3.2. Relationship between the Hydrological Cycle and the TWS

      Considering the effects of precipitation, evapotranspiration, and runoff on the TWS over the Tibetan Plateau, the runoff coefficient was calculated and the proportions of evapotranspiration, runoff, and TWSC in the corresponding precipitation were counted at the 12 sub-regions (Figure 9). As a whole, precipitation over the Tibetan Plateau was mainly lost through evapotranspiration, especially in the Yellow River source region (94.25%), Qaidam Basin (93.02%), Inland River region (92.72%), the Yangtze River source region (87.22%), and the Qinghai Lake region (85.76%). The corresponding runoff proportion in the northern part of the Tibetan Plateau was limited, and the runoff coefficient was mainly lower than 0.2. In the Yarlung Zangbo River Basin and Himalayas, runoff accounted for about 50% of the precipitation and the runoff coefficient was about 0.5. Influenced by snow and glacier melting, the runoff coefficient in the Karakoram Mountains of the northwestern plateau, the Nyainqin-Tanglha Mountains in the northeastern Yarlung Zangbo River Basin and the southern Himalayas were greater than 1.0.

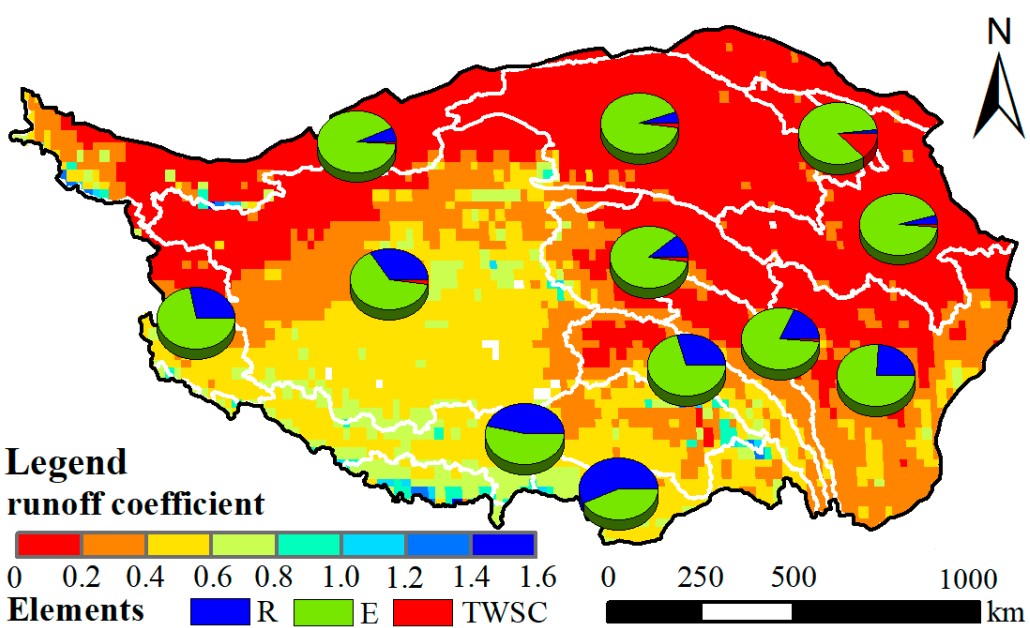

**Figure 9.** Regional statistical diagram of the proportions of each hydrological cycle's elements in precipitation and spatial distribution of runoff coefficient over the Tibetan Plateau.

## 5. Discussion

### 5.1. Climate Change and Water Resources Security

Climate change can alter the large-scale atmospheric circulation through physical and chemical interactions between different layers, such as ocean–atmosphere and land–atmosphere interactions. The changes in precipitation, melting snow, and glacier influence water resources and water quality, which in turn have wide-ranging impacts on natural ecosystems and human societies around the world. In most areas of the Tibetan Plateau, soil moisture is the important component of water resources and occupies a percentage of more than 90% in terrestrial water storage. Climate change influences soil moisture by interfering with the processes of precipitation infiltration, land surface evaporation, and plant transpiration. The relationships between climate change and soil moisture are of great significances for water resources distribution and security over the plateau, which were explored through the temperature and precipitation from the CMFD and 0–200 cm soil moisture during the period 1981–2015 (Figure 10). Affected by precipitation, rainfall intensity, rainfall duration, and soil properties, the increase in precipitation was accompanied by a rising in soil moisture, which was significantly positively correlated in the eastern Tibetan Plateau, while the soil moisture and precipitation showed a negative correlation in most areas of the Qiang-tang Plateau, the Indus River and Yarlung Zangbo River source regions, and the Himalayas. Considering the effects of temperature, the soil moisture in the Yangtze River, Yellow River, and Lancang River source regions showed an obviously positive correlation with temperature, and the correlation coefficients were above 0.6, which means that the rise in temperature in the high-altitude areas may cause glacial and permafrost melting, and then the soil moisture increased. In the northeastern part of the Qiang-tang Plateau, which are connected to the Qaidam Basin and Inland River region, the rising temperature could make the evapotranspiration increase and the soil moisture decrease. The correlation between soil moisture and temperature in other areas of the Tibetan Plateau was insignificant, and the correlation coefficients were mainly between −0.2 and 0.2.

The influences of climate change on soil moisture were also analyzed at an interannual scale. The annual temperature, precipitation, and soil moisture were jointly compared in the 12 sub-regions from 1981 to 2015 (Figure 11), and the changing trend of each element was expressed obviously through the calculated anomalies. In the Inland River region (a), Qaidam Basin (l), and Qinghai Lake region (k), soil moisture and precipitation showed a

basically consistent trend, which was both precipitation and soil moisture continuing to rise during the period 1981–2015, and the average value after 2000 was significantly higher than that before 2000. The Yangtze River (h) and Yellow River (j) source regions, Nujiang River Basin (f), and Lancang River Basin (g) belong to the monsoon climate region, where the relationships between precipitation and soil moisture were very consistent, negative anomalies in soil moisture were often accompanied by lower temperature and precipitation than the multi-year average values, and vice versa. In the Qiang-tang Plateau (b), Indus River Basin (c), and the Himalayas (e), as the temperature increased, the precipitation increased, and the soil moisture decrease may be due to the topographic and vegetation conditions.

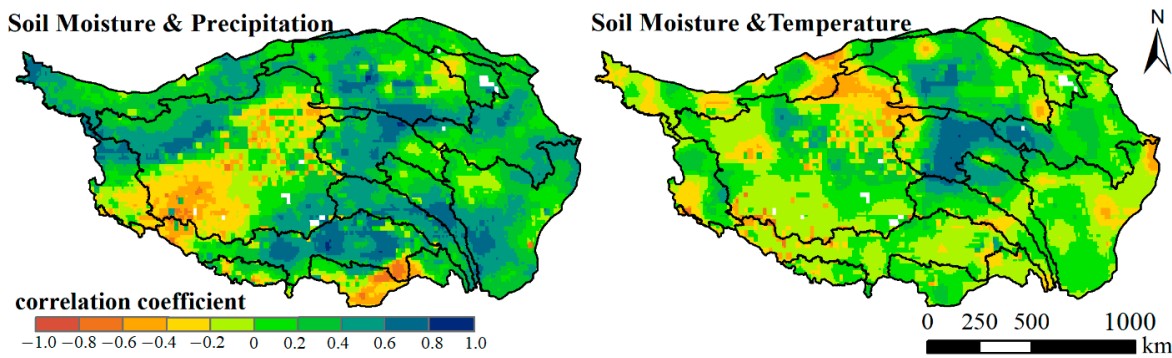

**Figure 10.** Spatial distributions of correlation coefficients between soil moisture and precipitation, temperature.

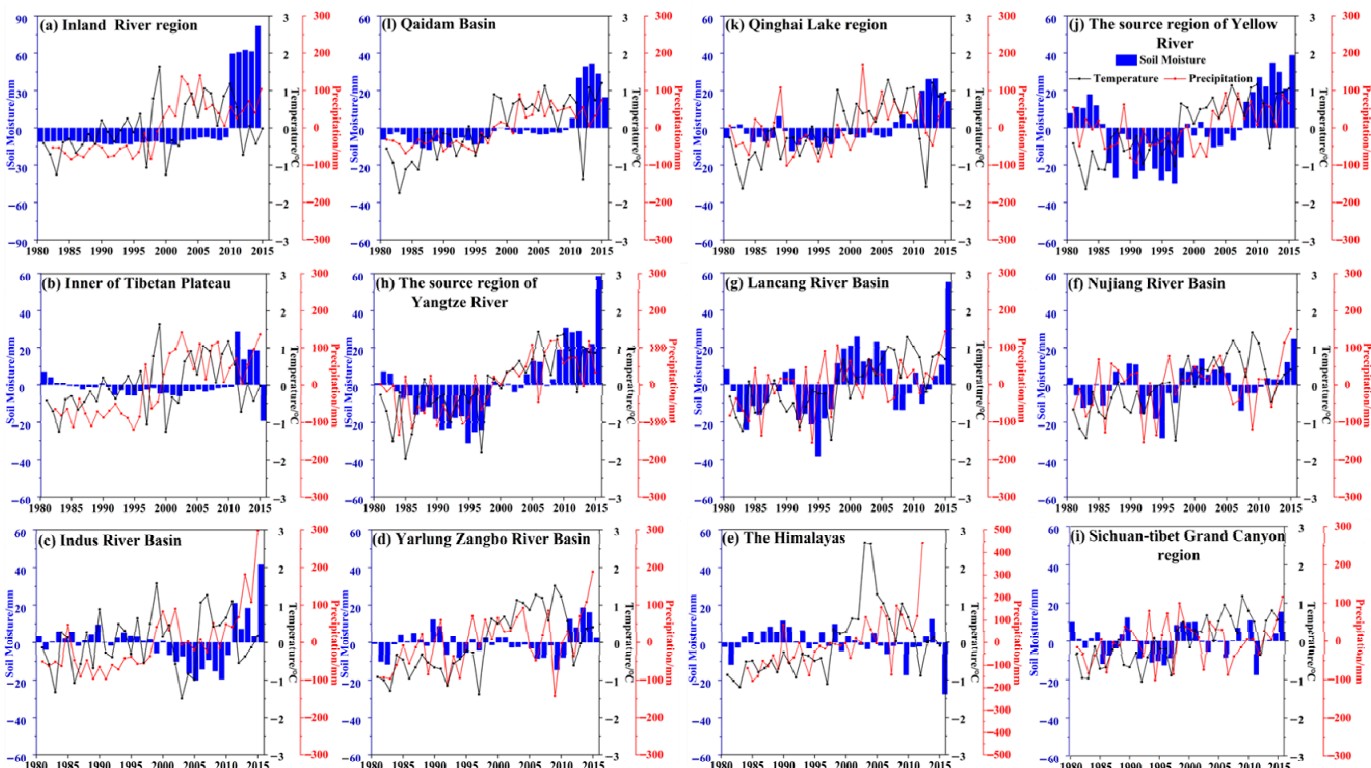

**Figure 11.** The annual temperature, precipitation, and soil moisture anomalies for each sub-region over the Tibetan Plateau during the period 1981–2015.

## 5.2. Water Conservation Capacity of Vegetation over the Tibetan Plateau

Vegetation is an important component of terrestrial ecosystems and is a bond of connecting various elements such as the atmosphere, soil, and hydrology. Water resources are a decisively driving factor of vegetation dynamics; about 61% of global vegetation cover is primarily controlled by water conditions [66], on the other hand, spatiotemporal variations

in vegetation also can also cause changes in parameters, affecting the balances of surface energy and the water cycle. Precipitation is a common indicator to investigate how the water conditions influence vegetation. Previous studies have investigated the responses of vegetation to water conditions, mainly relying on the surficial and indirect information from precipitation [66–68]; the influences of terrestrial water storage on vegetation have been gradually explored in recent years. The satellite-based normalized difference vegetation index (NDVI) could indicate the growth, spatial distribution, and change trend of vegetation. The relationship of NDVI with terrestrial water storage showed the water conservation capacity of vegetation over the Tibetan Plateau during the period 1981–2015 (Figure 12), where the blank areas did not pass the 95% significance test. The NDVI over the western Tibetan Plateau showed the characteristics of low and increasing values; on the other hand, the eastern Plateau was high but showed decreasing values. The positive correlation was relatively strong between NDVI and terrestrial water storage in the Qinghai Lake region (0.62 **), the inland river region (0.49 **), the Qaidam Basin (0.46 **), and the Yellow River source region (0.41 **). Moreover, both the NDVI and terrestrial water storage presented a significantly increasing trend in the north, accounting for about 13.20% of the plateau, indicating that the water conservation capacity has been improved to a certain extent during the process of the continuous improvement of vegetation in the north. In the source regions, NDVI and terrestrial water storage were mainly in an inverse relationship, whereas in the Yangtze River, Yellow River, and Lancang River source regions, the NDVI has decreased and terrestrial water storage has increased in recent years. While in the Indus River and Yarlung Zangbo River source regions, the NDVI increased while terrestrial water storage decreased. The increase in the NDVI leads to a greater rise in water consumption for vegetation growth and then a decline in terrestrial water storage probably due to the high altitude where the source areas are located. The vegetation growth in the southeast, especially the tropical monsoon climate zone around the Yarlung Zangbo River Grand Canyon and the Himalayas, was generally good but has had a certain degradation trend in recent years, and terrestrial water storage has also declined to a certain extent.

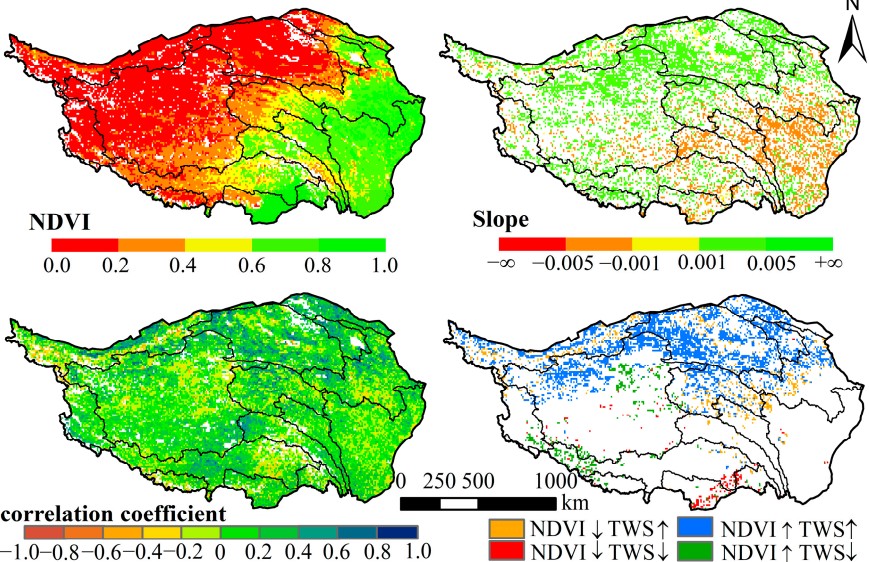

**Figure 12.** Characteristics of the underlying surface and its relationship with terrestrial water storage over the Tibetan Plateau during the period 1981–2015.

In addition, the temporal changing trends of the NDVI and terrestrial water storage anomalies from 1981 to 2015 were detected in the 12 sub-regions (Figure 13). In the Inland River Basin (a), Qaidam Basin (l), and Qinghai Lake region (k), where the terrestrial water storage was relatively low and the evapotranspiration water consumption of vegetation accounted for a large amount in precipitation, both the terrestrial water storage and NDVI showed a fluctuating increase. When the terrestrial water storage had a high value, the

NDVI anomalies were also larger during the period 2011–2015. In the Yangtze River (h) and Yellow River (j) source regions, both the terrestrial water storage and NDVI tended to firstly decrease and then increase, and the continuous decrease in terrestrial water storage had certain cumulative effects on vegetation growth. Therefore, the NDVI minimum appeared later than the lowest terrestrial water storage. Nevertheless, when terrestrial water storage increased, the vegetation growth also improved accordingly. In other regions, the volatilities in terrestrial water storage and the NDVI were relatively strong and the variations were not significant, so no obvious correlations between them were detected in the temporal changing trend.

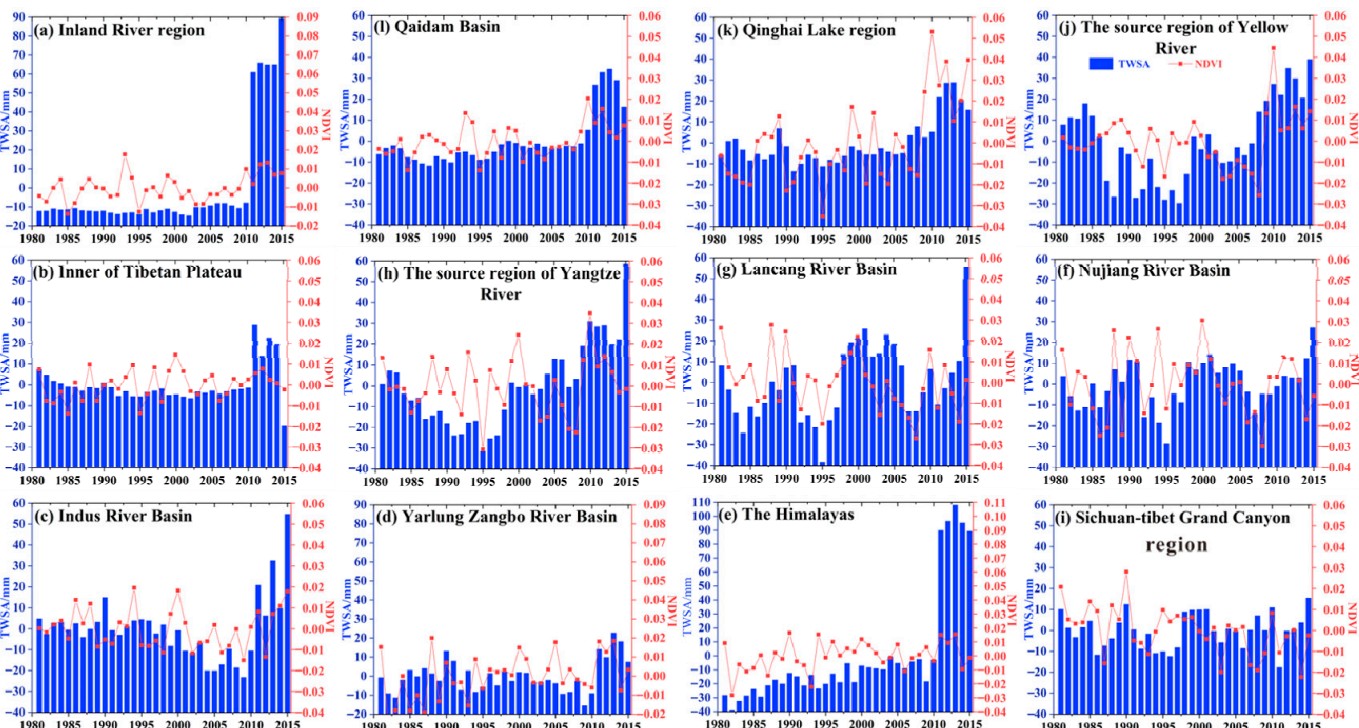

**Figure 13.** Temporal trend in terrestrial water storage anomaly and NDVI for each sub-region over the Tibetan Plateau during the period 1981–2015.

*5.3. Contribution, Uncertainty Analysis and Limitations of Current Study*

Investigating the spatio-temporal evolutions and corresponding relationships of vegetation cover, water, and heat resources over the Tibetan Plateau is of great significance for guiding local production practices, such as water resources management and agricultural development. The statistical results on the average values of TWS and the NDVI on the municipal administrative divisions scale during the period of 1981–2015 are shown in Table 4. The TWS presented the lowest values of about 300 mm in the Tian shui, Ding Xi, Jiuquan, and Zhangye of Gansu Province, while the values exceeding 500 mm were mainly distributed in the provinces of Tibet, Yunnan, and Sichuan. In Xinjiang, Dingxi, and Jinchang of Gansu Province and the western parts of Tibet, the NDVI was below 0.3. There is a certain correlation between water resources and vegetation cover in the mountainous areas, but it is not absolute [69–73]. The results are similar to the others overall and have greater spatial heterogeneity over the Tibetan Plateau than the mountainous areas of the Taihang Mountains, the Hengduan Mountains, and the Guizhou-Guangxi Karst than those from [73]. The research is conducive to public policies and decision-making mainly for the comprehensive regulation of resource allocation and the promotion of economic and social development.

**Table 4.** The statistics of the average TWS and NDVI over the Tibetan Plateau.

| Province | City/Region | NDVI | TWS/mm | Province | City/Region | NDVI | TWS/mm |
|---|---|---|---|---|---|---|---|
| Xinjiang | Kunyu | 0.11 | 336.86 | Gansu | Jinchang | 0.14 | 475.64 |
| | Kashgar Region | 0.34 | 340.15 | | Hui nationality of Linxia | 0.78 | 484.91 |
| | Kizilsuk | 0.10 | 373.65 | | Longnan | 0.74 | 491.72 |
| | Hotan Region | 0.15 | 418.12 | | Nyingchi | 0.60 | 526.89 |
| | Babingolemont | 0.17 | 421.54 | | Shigatse | 0.50 | 527.45 |
| Qinghai | Tibetan of Huangnan | 0.60 | 290.98 | | Lhasa | 0.68 | 538.46 |
| | Haidong | 0.69 | 309.20 | Tibet | Ali Region | 0.27 | 540.11 |
| | Tibetan of Hainan | 0.74 | 349.08 | | Shannan | 0.72 | 546.74 |
| | Tibetan of Haibei | 0.57 | 366.52 | | Nagqu | 0.64 | 565.96 |
| | Tibetan of Guoluo | 0.51 | 370.43 | | Qamdo | 0.26 | 608.67 |
| | Xining | 0.79 | 428.01 | Yunnan | Lisu nationality of Nujiang | 0.68 | 498.59 |
| | Mongolian of Haixi | 0.43 | 430.65 | | Lijiang | 0.72 | 519.74 |
| | Tibetan of Yushu | 0.62 | 436.95 | | Tibetan of Diqing | 0.82 | 541.93 |
| Gansu | Tianshui | 0.46 | 304.31 | Sichuan | Yi nationality of Liangshan | 0.70 | 499.17 |
| | Dingxi | 0.13 | 304.98 | | Tibetan of Ganzi | 0.79 | 509.76 |
| | Jiuquan | 0.39 | 310.58 | | Ya an | 0.87 | 535.93 |
| | Zhangye | 0.54 | 322.19 | | Tibetan of Aba | 0.82 | 553.87 |
| | Tibetan of Gannan | 0.70 | 392.12 | | Mianyang | 0.68 | 662.86 |

An increasing number of studies have used the GRACE dataset to analyze the spatiotemporal variabilities of terrestrial water storage around the world; however, the GRACE data have a quite rough resolution and some truncation, measurement, and leakage errors exist [74]. Some researchers have begun to compare the terrestrial water storage calculated by the water balance equation with the GRACE dataset [26]. In the Tianshan Mountains, comparing the data derived by the GRACE and GLDAS, it was found that there exists excellent consistency and significant liner relations, showing a decreasing trend due to the steady shrinkage of the Tianshan Mountain glaciers during the period 2003–2013; the monthly maximum occurred in April, while October contributed to the minimum [37]. In the Tibetan Plateau, Meng et al analyzed spatiotemporal terrestrial water storage changes through GRACE and the VIC-glacier model during the period 2003–2014; obvious increments were detected in the central and northern plateau and a distinct decrease could be found in the southern and northwestern plateau [14]. Precipitation and evapotranspiration mostly determine terrestrial water storage from the angle of water balance, and the changes in soil moisture and glacier mass are the main reasons in terms of hydrological components. Wang et al also concluded that in southern the Tibetan Plateau, where it was warm and wet, terrestrial water storage showed a significant decline, and then the soil moisture decreased and the groundwater storage and runoff increased. Otherwise, in the cold and dry northern Tibetan Plateau, the accumulation of terrestrial water storage could be attributed to increased soil moisture during the period 1992–2015 [75]. Dimri et al pointed to decreased precipitation and amplified warming signals over the Indian Himalayan region [76] and many other pieces of research have proven this [77–80]. Pepin et al also concentrated on the temperature and precipitation changes in the last 40 years over the mountains of the world and they found that stores of mountain snow and ice may be declining faster than previously assumed due to enhanced mountain warming [81]. Precipitation in the Himalayas has decreased since 1960 and the decreasing rate has intensified since 1990, while a continuous increasing trend was observed in the entirety of the Tibetan Plateau [79–81], which is consistent with the increasing total TWS of the plateau, but precipitation significantly decreased in the Himalayas during the period 1981–2015 in this study. Because of the uncertainty of the GLDAS-Noah data itself and the differences between the selected periods, the corresponding results are basically consistent with previous research, while differences also exist. The terrestrial water storage of the lake region in this study is null due to a lack of data. In addition, then, the terrestrial water storage in the Tibetan Plateau

northernly increasing and southernly decreasing overall, and the evapotranspiration in the north accounted for the majority of the water consumption. Apart from the Pamir Plateau, the Nyainqen-Tanggula Mountains, and the vicinities of Mount Everest, most of the proportion of soil moisture was more than 90%. In addition, terrestrial water storage in the southern Tibetan Plateau fluctuated to a greater extent than that in the north and the maximum value reached in July in the north and August in the south, which was affected by the monsoon climate.

## 6. Conclusions

In this study, a non-parametric Sen's slope and the Mann–Kendall trend test were applied to investigate the inter-annual and intra-annual spatiotemporal variabilities of the terrestrial water storage (TWS) at different spatial scales over the Tibetan Plateau during the period 1981–2015. Next, the water balance method combined with the geostatistical method were jointly used to analyze the components of TWS and explore the changing mechanism. In addition, the impacts of climate and vegetation changes on water resources and the water conservation capacity of vegetation were discussed in twelve sub-regions. The main findings can be summarized as follows:

(1) The TWS of the whole Tibetan Plateau increased at the speed of 0.7 mm/yr and gradually increased from north to south during the period 1981–2015. In most of the areas, the TWS value was between 300 mm and 600 mm. In the northern Tibetan Plateau, the TWS was low and characterized by stability within the year and obvious accumulation in the interannual scale. While in the south of the Tibetan Plateau, the high and decreased values distributed with apparently intra-annual fluctuations.

(2) In most areas, the TWS mainly consisted of soil moisture, which was 0–200 cm underground and occupied a percentage of more than 90% in the TWS. The plant canopy surface water increased from northwest to southeast, only accounting for 0–0.04% of the TWS. The soil moisture had the similar changing trend with the TWS, which increased in the north and decreased in the southern regions. Additionally, in the regions near Mount Everest, the proportion of snow water equivalent in the TWS can reach up to 98.22%.

(3) The precipitation, evapotranspiration, and runoff over the Tibetan Plateau had obviously spatial heterogeneity and gradually increased from north to south. The precipitation over the northern and northeastern Tibetan Plateau was mainly lost through evapotranspiration with the runoff coefficient lower than 0.2, while in the Himalayas, northeastern Yarlung Zangbo River Basin, and the northwest corner of the Tibetan Plateau, the runoff coefficients were larger than 1.0 due to the influence of snow melting.

(4) Apart from the positive correlation between soil moisture and temperature in the Yangtze River, Yellow River, and Lancang River source regions, the soil moisture in other regions of the Tibetan Plateau was more affected by precipitation than temperature. Both the NDVI and TWS presented a significantly increasing trend in the northern Tibet Plateau, while in the source regions of the rivers over the Tibet Plateau, such as the Yangtze River, Yellow River, Lancang River, Indus River, and Yarlung Zangbo River, the NDVI and TWS always showed opposite changing characteristics.

**Author Contributions:** Conceptualization, D.Z.; Methodology, Y.H. and D.Z.; Software, Y.H.; Validation, Y.H.; Formal analysis, Z.X. and G.W.; Data curation, Y.H.; Writing–original draft, Y.H. and D.Z.; Writing–review & editing, Z.X., G.W., D.P., B.P. and H.Y.; Visualization, Y.H.; Supervision, Z.X.; Project administration, D.Z. and G.W.; Funding acquisition, D.Z. All authors have read and agreed to the published version of the manuscript.

**Funding:** This study is jointly supported by the National Key Research and Development Program of China (Grant No. 2021YFC3201104 and No. 2021YFC3201502), Joint Open Research Fund Program of State key Laboratory of Hydroscience and Engineering and Tsinghua—Ningxia Yinchuan Joint

Institute of Internet of Waters on Digital Water Governance (Grant No. sklhse-2022-Iow07), and the Natural Science Foundation of Beijing Municipality (Grant No. 8202030).

**Data Availability Statement:** The authors would like to thank the Goddard Earth Science Data and Information for supporting the Global Land Data Assimilation System (GLDAS), the National Tibetan Plateau Data Center which provided the China Meteorological Forcing Dataset (CMFD), the Resource and Environ-mental Science Data Center of the Chinese for providing the Digital Elevation Model, the Institute of Soil Science, Chinese Academy of Sciences for providing the Soil Database of China, University of Maryland for providing the Land Cover Classification and National Aeronautics and Space Administration for providing the GIMMS NDVI3.

**Conflicts of Interest:** The authors declare no conflict of interest.

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
