# Peer review of "Attributing the Impacts of Vegetation and Climate Changes on the Spatial Heterogeneity of Terrestrial Water Storage over the Tibetan Plateau"

_remotesensing, doi:10.3390/rs15010117_

Round 1

Reviewer 1 Report

Reconsider after major revisions

Reviewer 2 Report

In the manuscript “Attributing the impacts of vegetation and climate changes on the spatial heterogeneity of terrestrial water storage over the Tibetan Plateau” authors determine the TWS analyzing several databases such as GLDAS, CMFD and GIMMS NDVI3g from 1981 to 2015. In this sense, using Man-Kendall trends and Sen´s slopes they determined inter-annual and intra-annual variations, finding a gradual increase of 0,7mm/year of TWS throughout the Tibetan Plateau. Additionally, authors identified the changing mechanisms from a component analysis and a water cycle analysis point of view, concluding that the soil humidity is in this case the main factor, representing more than 90% of TWS. According to the results, precipitation is mainly lost by evapotranspiration in most of the Tibetan Plateau Area. Finally, water conservation capacity of vegetation on the area is assessed. The manuscript results are valuable and have relevant implications for public policies and decision making.

However, I have some comments and questions, suggesting they must be specified in the manuscript.

QUESTION #1 Do you consider that TWS change trends and the average slope of the TWS over the Tibetan Plateau are directly related to eastern monsoon climate? What has been the main change in the Eastern Monsoon climate behavior during the analyzed period (1981-2015) over the Tibetan Plateau?

QUESTION #2. It would be important to add a conclusion related to the direct contribution of this study with public policies and decisions making over the Tibetan Plateau.

QUESTION #3. Please, consider mentioning the study area total surface in km2.

QUESTION #4. In figures 5 and 6 at different soil depths the different scales used could conduce to misunderstanding. For a better interpretation, the same scale must be used.

QUESTION #5. It would be useful to include a soil cover/use and a soil type spatial representation to enhance the results understanding.

QUESTION #6. No details can be seen in Figures 10 and 12, please correct.

Round 2

Reviewer 1 Report

 Accept in present form

Reviewer 3 Report

I want to thank the authors for their insightful corrections to the paper. they have been able to incorporate changes to reflect most of the suggestions provided by the reviewer. The paper can be now accepted for publication in Remote Sensing (MDPI).